# DataSP: A Differential All-to-All Shortest Path Algorithm for Learning Costs and Predicting Paths with Context

**Alan A. Lahoud**[1]  **Erik Schaffernicht**[1]  **Johannes A. Stork**[1]

[1]Center for Applied Autonomous Sensor Systems (AASS), Örebro University, Örebro, Sweden

## Abstract

Learning latent costs of transitions on graphs from trajectories demonstrations under various contextual features is challenging but useful for path planning. Yet, existing methods either oversimplify cost assumptions or scale poorly with the number of observed trajectories. This paper introduces DataSP, a differentiable all-to-all shortest path algorithm to facilitate learning latent costs from trajectories. It allows to learn from a large number of trajectories in each learning step without additional computation. Complex latent cost functions from contextual features can be represented in the algorithm through a neural network approximation. We further propose a method to sample paths from DataSP in order to mimic observed paths' distributions. We prove that the inferred distribution follows the maximum entropy principle. We show that DataSP outperforms state-of-the-art differentiable combinatorial solver and classical machine learning approaches in predicting paths on graphs. The code is available at `https://github.com/AlanLahoud/dataSP`.

## 1 INTRODUCTION

Learning routes from demonstrations conditioned on contextual features plays a crucial role in traffic management and urban planning [Hirakawa et al., 2018, Rossi et al., 2019, Gao et al., 2018]. The underlying assumption is that agents performing these demonstrations attempt to optimize a latent cost when trying to reach a destination [Ziebart et al., 2008a, Finn et al., 2016]. These costs might balance variables such as trip duration, comfort, toll prices, and distance.

Recovering these latent costs not only allows to model the underlying decision-making process but also to improve traffic flow through the anticipation of congestion [Xiong

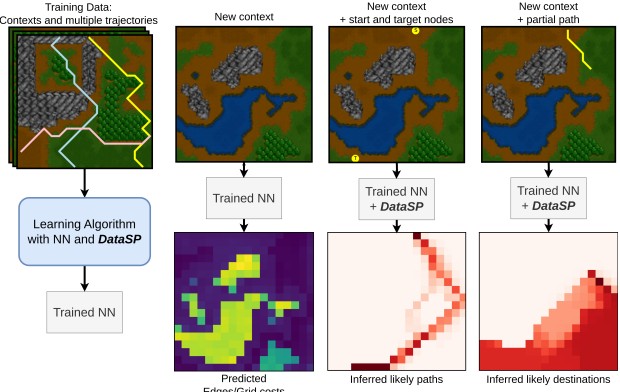

Figure 1: DataSP facilitates the process of learning from demonstrations, usually expert's paths, and contexts like images. The bottom row shows outputs of our model. The trained NN infers latent costs, where dark blue is less costly. DataSP is then used to infer likely paths and destinations, where darker red indicates a higher likelihood.

et al., 2018], route prediction [Froehlich and Krumm, 2008], the provision of real-time navigational guidance to drivers [Liang and Wakahara, 2014], and the facilitation of targeted advertising strategies [Li et al., 2016].

Inverse reinforcement learning [Ziebart et al., 2008a,b, Nguyen et al., 2015], which focuses on learning the costs associated with edges or transitions from observed trajectories, is one popular way to approach the problem. However, these methods generally assume a linear latent cost to simplify the learning process.

Recent approaches use neural networks (NNs) with combinatorial solvers to learn from contextual features and combinatorial solutions in an end-to-end manner. Pogančić et al. [2020] apply these to route prediction, assuming trajectories are optimal and using combinatorial solver blocks as shortest path solvers. Despite their innovation, these methods struggle to learn from a large number of trajectories since they typically need to solve a shortest path problem for each

observed trajectory, which does not scale favorably.

In this paper, we propose a novel method for learning latent costs from observed trajectories by encoding these trajectories into frequencies of observed shortcuts. We achieve this by differentiating through the Floyd-Warshall (FW) algorithm, leveraging its ability to solve all-to-all shortest path problems in a single run based on shortcuts. This approach enables us to learn from a large volume of trajectories with various starting and target graph nodes. In a single step of the learning process of a NN, we can capture a substantial amount of information about the latent costs within the graph structure.

When attempting to differentiate through FW, there are two issues to address. The first problem is that the gradients computed from FW's path solutions with respect to its inputs are non-informative (zero almost everywhere), due to their combinatorial nature [Abbas and Swoboda, 2021]. The second issue arises because FW provides an exact solution, implying an expectation for demonstrations to be optimal. This assumption is unrealistic for data obtained from human demonstrations, as observed in Ziebart et al. [2008a]. Although the agents producing these trajectories attempt to optimize paths according to latent costs, we assume observed trajectories are suboptimal due to agents' imperfect behavior.

We address both issues by introducing DataSP, a Differentiable all-to-all Shortest Path algorithm, which serves as a probabilistic and differentiable adaptation of the FW algorithm. By incorporating smooth max and argmax operators [Nesterov, 2005, Niculae and Blondel, 2017] into DataSP, we relax the solution to allow for informative backpropagation. Our proposed method is valuable not only for learning latent costs but also for predicting likely trajectories. We demonstrate that the output of DataSP is effective for predicting trajectories between various fixed start and end nodes across the graph structure in new contexts, and for inferring likely destinations or future nodes given a partial path in a new context. Figure 1 showcases some of our results, highlighting the capabilities of our method.

## 1.1 OUR CONTRIBUTIONS

- We propose a differentiable and probabilistic all-to-all shortest path algorithm that utilizes smooth approximations for the min and argmin operators, enabling backpropagation through shortest paths computation. This allows connecting neural network architectures to DataSP to learn non-linear representations of latent edges' costs based on contextual features.

- To learn from a substantial amount of information across various trajectories in each learning step, we propose a loss function that computes the dissimilarity between observed and inferred shortcut frequencies.

- Inferred edges' costs are utilized by the DataSP algorithm to generate paths that mimic the observed behavior, demonstrating that DataSP is not only useful for learning but also to infer paths. We prove that the probability distribution of the *total cost* of the generated paths follows the maximum entropy principle.

## 1.2 RELATED WORK

Studies in Inverse Reinforcement Learning (IRL) have focused on estimating latent costs from suboptimal demonstrations, primarily through the alignment of feature frequencies across transitions [Ziebart et al., 2008a,b, Henry et al., 2010, Vasquez et al., 2014, Nguyen et al., 2015]. These approaches commonly presuppose linear models for latent costs, limiting their flexibility and applicability to complex scenarios. Finn et al. [2016] introduced the use of nonlinear cost functions articulated through NNs, albeit within the confines of a continuous trajectory space applied to robotics tasks.

Expanding on this foundation, the domain of Deep Inverse Reinforcement Learning has seen advancements through works such as Wulfmeier et al. [2017], Fernando et al. [2020], which incorporate NNs to estimate latent costs. Despite these innovations, such approaches rely on estimating the gradients based on state visitation frequencies.

Works such as Amos and Kolter [2017], Mensch and Blondel [2018], Agrawal et al. [2019], Pogančić et al. [2020] focus on computing gradients (or an approximation of the gradients) of optimal solutions for various types of mathematical programming, allowing to use NNs to learn latent variables from demonstrations in an end-to-end manner with an optimization solver component in the loop. This allows implementations to map context features to optimal observed demonstrations with automatic differentiation.

Pogančić et al. [2020] propose learning latent costs from optimal shortest path solutions by approximating gradient computations of combinatorial solutions. Berthet et al. [2020] address similar problems by introducing random noise and utilizing Monte-Carlo methods for gradient computations, enabling learning from both optimal and suboptimal solutions. However, these methods are applied to single source-target trajectories and require to call optimization solvers proportionally to the amount of observed trajectories. As this implies steep computational costs, our method is instead designed to learn from multiple trajectories in each iteration of the learning process, allowing to have more informative learning steps during the NN optimization process.

## 1.3 PROBLEM FORMULATION

Here, we formalize the problem of learning the latent costs from observed trajectories and contextual features. First, we define the data availability assumption and then formulate

the minimization problem to guide the learning process.

**Data.** We formulate the problem within a given graph structure $G^{|V|} = (V, E)$, where the nodes and existing edges between them are predetermined. The nodes in $V$ are sorted in ascending order from 0 to $|V| - 1$. The edge costs $\boldsymbol{y}$ are unknown and considered latent variables. We assume we observe $c$ optimal or suboptimal trajectories $\tau_{n,c}$ from various start and target nodes, for each respective context sample $x_n$, represented by features that might influence the latent costs and trajectory choices. These features can be represented by tabular data containing variables such as time of the day, weather, or images. The training data is written as $\mathcal{D} = \{\boldsymbol{x}_n, \tau_{n,c}\}_{n=1}^N$. Each trajectory is represented as a sequence of nodes in the graph, i.e., integer values.

**Minimization Problem.** Let $F_{n,c}$ be a variable that encodes all the $c$ observed trajectories within the context $\boldsymbol{x}_n$. Let $P_n^\omega$ be a variable that encodes suboptimal trajectory information inferred from predicted (latent) costs $\hat{\boldsymbol{y}}_n^\omega := \hat{\boldsymbol{y}}^\omega(\boldsymbol{x}_n)$, where $\omega$ are NN weights. We aim to minimize the following empirical loss:

$$\omega^* = \arg\min_{\omega \in \Omega} \frac{1}{N} \sum_{n=1}^N \left[ \mathcal{L}_S(P_n^\omega, F_{n,c}) + \alpha \mathcal{L}_P(\hat{\boldsymbol{y}}_n^\omega, \boldsymbol{y}^p) \right]. \tag{1}$$

We define both variables $P$ and $F$ in the next section, revisiting the loss function $\mathcal{L}_S$ in more detail. If prior knowledge about the edges' cost $\boldsymbol{y}^p$ is available, e.g., euclidean distance between nodes, the loss is regularized by $\mathcal{L}_P$, where $\alpha$ is a hyperparameter to set the regularization factor. Solving this minimization means that the NN is able to provide useful edges' costs, i.e., costs that can be used to mimic observed trajectories, while keeping the values close to $\boldsymbol{y}^p$.

## 2 PRELIMINARIES

This section provides important background that we use in our proposed DataSP algorithm. It includes a brief review of the classical FW algorithm, and covers techniques for enabling informative gradient flow through smooth max and argmax operators.

### 2.1 FLOYD-WARSHALL ALGORITHM

FW [Floyd, 1962] is a Dynamic Programming [Bellman, 1966] method used to find the shortest paths between all pairs of vertices in a graph $G^{|V|} = (V, E)$, where $V$ and $E$ are sets representing nodes and edges, respectively. The input is a cost matrix $M \in \mathbb{R}^{|V| \times |V|}$ representing the cost of edges between adjacent nodes in the graph $G^{|V|}$, which could signify distance, time, or other metrics. The algorithm iteratively considers each node $k$ as a potential intermediate node in the paths between all node pairs, as detailed in Algorithm 1. The three nested loops lead to a time complexity of

---

**Algorithm 1** Floyd-Warshall Algorithm

---
**Input:** A cost matrix $M \in \mathbb{R}_+^{|V| \times |V|}$
**Output:** The optimal distance matrix $M$ and predecessor matrix $R$

1: Initialize $R$ based on $M$
2: **for** $k = 0$ to $|V| - 1$ **do**
3:     **for** $i = 0$ to $|V| - 1$ **do**
4:         **for** $j = 0$ to $|V| - 1$ **do**
5:             $M[i,j] \leftarrow \min(M[i,k] + M[k,j], M[i,j])$
6:             **if** $M[i,k] + M[k,j] < M[i,j]$ **then**
7:                 $R[i,j] \leftarrow R[k,j]$
8:             **end if**
9:         **end for**
10:     **end for**
11: **end for**
12: **return** $M, R$

---

$\mathcal{O}(|V|^3)$. During each iteration, it checks whether the path between two vertices $i$ and $j$ can be shortened by including an intermediate vertex $k$. If so, it updates the path length in the matrix with $M[i,j] \leftarrow M[i,k] + M[k,j]$. When the algorithm is finished, $M[i,j]$ contains the optimal cost between $i$ and $j$ for all pairs. FW also stores the predecessor matrix $R$ that is used to track predecessor nodes to recover the optimal path between two given nodes.

### 2.2 SMOOTH MAX OPERATORS

A particular and important approximation to the max operator that we leverage in this work is represented by the *logsumexp* function, making its gradient, the *softmax* or *gibbs* function to serve as an approximation to the argmax operation [Nesterov, 2005, Niculae and Blondel, 2017]. In this paper, we use the *min* version of these smoothed operators as follows to simplify the notation in the following sections:

$$\min_\beta : \mathbb{R}^K \to \mathbb{R}, \qquad \min_\beta(\boldsymbol{v}) = -\frac{1}{\beta} \log \sum_{i=1}^K e^{-\beta v_i}, \tag{2}$$

$$\Phi_\beta : \mathbb{R}^K \to [0,1]^K, \qquad \Phi_\beta(\boldsymbol{v}) = \frac{e^{-\beta \boldsymbol{v}}}{\sum_{i=1}^K e^{-\beta v_i}}. \tag{3}$$

## 3 METHODS

After introducing the Differentiable all-to-all Shortest Path (DataSP) algorithm, we will include it in a learning loop to estimate latent costs, and how to sample paths from the model.

## 3.1 DIFFERENTIABLE ALL-TO-ALL SHORTEST PATHS

Our proposed gradient-informative variant of FW is presented in Algorithm 2. DataSP diverges from the classical FW in two key aspects. First, it sequentially updates the cost matrix $M$ with the smooth min operator $\min_\beta$. Second, it stores smooth argmin values denoted by the tensor $P$, our desired output of the algorithm.

**Theorem 1.** *Let $M^{(k)}$ denote the value of $M$ after the $k^{th}$ iteration of the DataSP loop. Let $G^k$ be the graph containing all nodes from 0 to $k-1$. Let $\boldsymbol{f^k}_{i\to j}$ be the vector containing the costs of all possible paths in the subgraph $G^k$ going from node $i$ to node $j$ (including $i$ and $j$). Then, it holds that: $M^k[i,j] = min_\beta(\boldsymbol{f^k}_{i\to j}) \quad \forall i,j,k \in V$.*

The theorem above shows that DataSP process keeps $M$ consistent to the smooth min operator of the path costs through all iterations. After DataSP is complete, $M$ contains the shortest distance matrix for all pairs of nodes in $G^{|V|}$ smoothed by $\beta$. The special case $\beta \to \infty$ is equivalent to the classical FW solution. The proof is detailed in Appendix B.

$P$ is an alternative way to represent paths in a probabilistic fashion. For the iteration $k$ in DataSP, $p_s$ evaluates the likelihood of $k$ to be included in the optimal path using the smooth argmin operator. In this specific iteration, if node $k$ is included in the optimal path, then $k$ represents the node with the highest index in the optimal path between $i$ and $j$ due to the ascending order of the outer loop, i.e., nodes $> k$ are not yet evaluated. Therefore, we interpret $P[i,j,k]$ as the probability that $k$ is the *highest intermediate node* between $i$ and $j$ in an optimal path. For the same iteration, $P[i,j,c]$ is normalized for all $c < k$ and $c = i$ using $p_d = 1 - p_s$, maintaining the previous probabilities proportion while reassuring that $\sum_{k=0}^{|V|-1} P[i,j,k] = 1$, making $P[i,j]$ to be a discrete probability distribution. We interpret $P[i,j,i]$ as the probability that there is no intermediate nodes when going from $i$ to $j$, and we initialize these values with 1, i.e., the initial guess is that the optimal path from $i$ to $j$ is a direct path.

**Theorem 2.** *Let $P^{(k)}$ denote the value of $P$ after the $k^{th}$ iteration of the DataSP loop. Let $G^k$ be the graph containing all nodes from 0 to $k-1$. Let $\boldsymbol{f^{k'}}_{i\to k'\to j}$ be a vector containing the costs of all paths in the subgraph $G^{k'}$ going from $i$ to $j$ through the node $k'$. Then, it holds that:*
$$P^k[i,j,k'] = \frac{\sum e^{-\beta \boldsymbol{f^{k'}}_{i\to k'\to j}}}{\sum e^{-\beta \boldsymbol{f^k}_{i\to j}}} \quad \forall i,j,k \in V \text{ and } k' <= k.$$

The theorem shows that $P$ can be expressed in terms of path costs, i.e., $\boldsymbol{f}$. These analytical values of $P$ help us to derive the likelihood of sampling a specific path based on its cost value, as we detail in Section 3.4. The proof is detailed in Appendix B.

---

**Algorithm 2** DataSP Algorithm

---
**Input:** A cost matrix $M \in \mathbb{R}_+^{|V|\times|V|}$
**Output:** Shortcut Probabilities $P \in [0,1]^{|V|\times|V|\times|V|}$

---

1: Init $P[i,j,k] = \begin{cases} 1 & \text{if } i=k \text{ and } M[i,j] < \infty, \\ 0 & \text{otherwise} \end{cases}$
2: **for** $k = 0$ to $|V| - 1$ **do**
3:    **for** $i = 0$ to $|V| - 1$ **do**
4:       **for** $j = 0$ to $|V| - 1$ **do**
5:          $(p_s, p_d) \leftarrow \Phi_\beta(M[i,k] + M[k,j], M[i,j])$
6:          $P[i,j,k] \leftarrow p_s$
7:          $P[i,j,c] \leftarrow p_d P[i,j,c] \quad \forall c < k \text{ and } c = i$
8:          $M[i,j] \leftarrow \min_\beta(M[i,k] + M[k,j], M[i,j])$
9:       **end for**
10:    **end for**
11: **end for**
12: **return** $P$

---

## 3.2 LEARNING WITH DATASP

In this subsection, we introduce the three foundational elements required to facilitate learning from suboptimal paths using DataSP: data preprocessing, the loss function, and the learning algorithm integrating a NN with DataSP.

**Encoding observed trajectories.** We process each set of $C$ trajectories into an observed *shortcut frequency tensor $F$*. We define $F \in [0,1]^{|V|\times|V|\times|V|}$ as the frequency of each node being the *highest intermediate node* between any two nodes in a dataset or in a batch, i.e., $F[i,j,k]$ is the number of times that $k$ is the *highest intermediate node* between $i$ and $j$ divided by the total observed paths (or subpaths) from $i$ to $j$. Note that $\sum_{k=0}^{|V|-1} F[i,j,k] = 1 \quad \forall i,j \in V$, with all values greater than 0, making $F[i,j]$ a discrete probability distribution for all $i,j$. The processed training data is then denoted as $\mathcal{D} = \{\boldsymbol{x}_n, F_{n,C}\}_{n=1}^N$.

**Shortcuts Loss Function ($\mathcal{L}_S$).** We propose the Shortcut Loss Function $\mathcal{L}_S$ as a dissimilarity measure computed between the *Shortcut Frequency Tensor $F$*, representing the distribution of shortcut decisions computed from the observed trajectories in the training data, and the predicted distribution $P$ of the *highest intermediate nodes* outputted from our algorithm DataSP. The comparison of $P$ and $F$ involves two tensors that encapsulate a large amount of path information in a single run of DataSP, offering a more informative feedback than methods attempting to align individual paths. When $P[i,j,:] \approx F[i,j,:]$ for all $i,j$, we approximately match their path reconstructions, allowing us to mimic the behavior of observed paths using $P$ in Section 3.4. Although many options of dissimilarity between $P$ and $F$ are possible, we chose to use the KL divergence in all of our experiments containing suboptimal observed paths due

to its convexity in the space of probability distributions. So we write the loss as

$$\mathcal{L}_S = \frac{1}{|D|} \sum_{(i,j) \in D} \sum_{k=1}^{K} F[i,j,k] \log \frac{F[i,j,k]}{P[i,j,k]}, \quad (4)$$

where $D$ is a set containing the combinations of source and target nodes of observed trajectories.

**Learning Algorithm.** We incorporate DataSP and the proposed loss function in Algorithm 3. The main motivation of this algorithm is to optimize the weights of the NN based on $\mathcal{L}_S$. By doing that, the NN is able to infer latent edges' costs that can reconstruct observed trajectories. Line 2 of the algorithm is the forward process of the NN to infer the unknown edges' costs. Line 3 fulfills the unknown elements of $M$, while the other elements are infinite due to lack of direct edges. Using DataSP, Line 4 computes the learned $P$ for the NN weights $\omega$ and the input sample $x_n$. Finally, line 5 updates the NN's weights following Equation 1. We use automatic differentiation for the backpropagation computation. It is noteworthy that increasing the number of selected trajectories $c$ for a single context does not affect the complexity of the learning algorithm.

---

**Algorithm 3** Learning with DataSP

**Components:** NN Parameters $\omega$, Learning Rate $\eta$
**Component:** A connected graph $G = (V, E)$
**Input:** Prior edges' cost $\boldsymbol{y}^p \in \mathbb{R}^{|E|}$
**Input:** Training Data $\{(\boldsymbol{x}_n, F_{n,c})\}_1^N$
**Output:** Optimized weights $\omega$

1: **for** $n = 1$ to $N$ **do**
2:     $\hat{\boldsymbol{y}}_n^\omega := \hat{\boldsymbol{y}}^\omega(\boldsymbol{x}_n)$ {forward pass of NN}
3:     Fill the cost matrix $M_n^\omega = M(\hat{\boldsymbol{y}}_n^\omega, G)$
4:     $P_n^\omega \leftarrow \textbf{DataSP}(M_n^\omega)$ {Algorithm 2}
5:     $\omega \leftarrow \omega - \eta \nabla \mathcal{L}_S(F_{n,c}, P_n^\omega) - \eta\alpha \nabla \mathcal{L}_P(\hat{\boldsymbol{y}}, \boldsymbol{y}^p)$
6: **end for**
7: **return** $\omega$

---

### 3.3 GRAPH SAMPLING

Like the classical FW algorithm DataSP's time complexity is $\mathcal{O}(|V|^3)$, which makes the whole learning process in Algorithm 3 slow. DataSP necessitates a significant memory allocation to manage the gradients associated with the shortcut probability tensor $P$. This means that the algorithm scales poorly with the number of nodes in the graph.

Besides minimizing redundant computations and vectorizing DataSP (detailed in Appendix A), we propose using a simple graph sampling technique during the training process to control memory usage in DataSP. During each learning iteration, we randomly sample nodes to be excluded. The exclusion process involves sequentially removing nodes and recalculating the connections of their neighboring nodes by

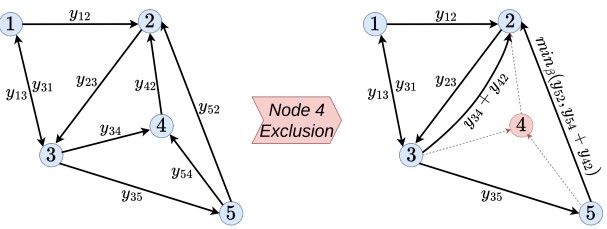

Figure 2: Example of one iteration of node exclusion in graph sampling: left shows the original graph; right shows exclusion of node 4 and its edges (in red). The remaining edges' costs are updated with local smooth min operations.

determining the shortest paths in a localized section of the graph solved using the smooth min operator. Solving this does not have a cubic dependency on the number of nodes in the graph. Instead, it depends on the number of edges connected to the excluded nodes. With each exclusion, the cost matrix's size is reduced by one element in each dimension. After a sequence of exclusions, it lowers the number of nodes to $|V_s| < |V|$, consequently lowering the memory demands for backpropagating through DataSP, as its input will be $M \in \mathbb{R}_+^{|V_s| \times |V_s|}$ and the output $P \in \mathbb{R}_+^{|V_s| \times |V_s| \times |V_s|}$. An illustrative example of a single iteration in the exclusion process is depicted in Figure 2.

By implementing this technique with the smooth min operator, we keep the elements of the resultant $M$ the same as it would be in the original graph for the set of the remaining nodes in $V_s$. This approach aligns with the characteristic of preserving shortest path distances when simplifying a graph, as done in [Ruan et al., 2011]. The general idea is to sample a different set of nodes in each iteration of the learning algorithm. This condenses different portions of the graph into smaller graphs throughout the learning process, such that in a long term, the integral information of the original graph is retained.

It is important to note that for each iteration, we adjust the loss function $\mathcal{L}_S$ to only consider the shortcuts of the sampled nodes. To achieve this, we also apply the node exclusion process in the observed paths from the training data. Specifically, we remove the same nodes from the training data paths that were excluded in the learning procedure, since we want the inferred shortcuts to be similar to the observed shortcuts even after compressing the original graph.

### 3.4 INFERENCE FROM DATASP

#### 3.4.1 Sampling path between nodes $i$ and $j$.

The expected path prediction between pairs of nodes $i$ and $j$ can be found using the predicted edge costs as an input to any standard SPP algorithm such as Dijkstra. However,

in order to approximately reproduce the distribution of the observed paths instead of predicting a single path, we propose sampling paths using Algorithm 4. This sampling algorithm uses the DataSP's output, $P$, computed from predicted edges' cost, i.e., $P^\omega$. The algorithm recursively samples the *highest intermediate node* $H$ between two nodes (line 2) in a subpath and concatenates it to the rest of the path. The recursion ends when a source node is sampled (lines 3 and 4) indicating a direct edge in the subpath. In lines 6 to 9 we update the values of $P$ based on the sampled $H$ using the Bayes' theorem. If $H$ is the *highest intermediate node* between $i$ and $j$, both intermediate nodes between $i$ and $H$ ($P_L$ relates to the left subpath), and between $H$ and $j$ ($P_R$ relates to the right subpath), must be smaller than $H$. This sampling algorithm can access paths with limited cycles. In practice, we remove the sampled walks containing cycles as we assume we do not observe cycles in the data. We show examples of this in Appendix C.

---

**Algorithm 4** Sampling Paths Between Nodes

**Input:** Shortcut Probabilities $P \in [0,1]^{|V| \times |V| \times |V|}$
**Input:** A start and end node $(i, j)$
**Output:** $\tau_{i \to j}$ as a seq. of nodes

1: **function** SampleH$(i, j, P)$
2:     $H \sim P[i, j, :]$
3:     **if** $H = i$
4:         **return** $i \to j$
5:     **else**
6:         $P_L \leftarrow P$ and $P_R \leftarrow P$
7:         $P_L[i, H, k] \leftarrow 0 \quad \forall k > H$ and $k \neq i$
8:         $P_R[H, j, k] \leftarrow 0 \quad \forall k > H$
9:         Normalize $P_L[i, H, :]$ and $P_R[H, j, :]$ to sum 1
10:         $\tau_{i \to H} \leftarrow$ SampleH$(i, H, P_L)$
11:         $\tau_{H \to j} \leftarrow$ SampleH$(H, j, P_R)$
12:     **return** Concat$(\tau_{i \to H}, \tau_{H \to j})$

---

**Theorem 3.** *The probability distribution of sampling paths $\tau_{i \to j}$ from Algorithm 4 is $\Phi_\beta(\boldsymbol{f}_{i \to j})$, where $\boldsymbol{f}_{i \to j}$ is a vector containing the costs of all possible DataSP paths going from $i$ to $j$ in the considered graph.*

The proof is found in Appendix B. The theorem shows that the likelihood of sampling a particular path with Algorithm 4 is proportional to the exponential of the negative cost associated with that path, matching the maximum entropy principle as detailed in the Inverse Reinforcement Learning literature [Ziebart et al., 2008a]. In Appendix C, Table 4, we show results of a Monte Carlo simulation computing the frequency of sampled paths in a simple graph to validate our finding.

### 3.4.2 Likelihood of future nodes given a partial path.

Given a partial path $\tau = n_1 \to n_2 \to ... \to n_K$ and $P^\omega$, we want to compute the probability that $n_x$ is a des-

| Experiments | W | W-M2M | W-M2M-N |
|---|---|---|---|
| # train images | 10000 | 350 | 350 |
| # paths/image | 1 | 30 | 30 |
| # total paths | 10000 | 10500 | 10500 |
| suboptimal $\tau$? | No | No | Yes |
| Results DBCS | **94.4** (0.4) | 82.4 (1.1) | 24.1 (0.8) |
| Results DataSP | 94.6 (0.3) | **92.2** (0.6) | **51.0** (3.0) |

Table 1: Percentage of inferred trajectories with optimal costs. Standard deviation between brackets are computed over five restarts.

tination for all $n_x \in V$. To simplify, we make $n_K$ to be the highest node index in the graph by swapping $n_K$ with $|V| - 1$ if $n_K \neq |V| - 1$ before running DataSP for inference. This is done by simply swapping the $M$ values (input of DataSP). By doing that, the calculation of this conditional probability is $\Pr(n_x | H_{\tilde{\tau}}[n_1, n_x] = n_K)$, where $H_{\tilde{\tau}}[n_1, n_x]$ is the *highest intermediate node* between $n_1$ and $n_x$ in a suboptimal path between them. Applying the Bayes' theorem turns the probability to be proportional to $\Pr(H_{\tilde{\tau}}[n_1, n_x] = n_K | n_x) \Pr(n_x)$, where the first factor is simply $P[n_1, n_x, n_K]$, and the second factor is a prior related to $n_x$ destination likelihood, which can be learned or given by some knowledge about the paths and the graph.

## 4 EXPERIMENTS

This section outlines three experiments. The first predicts Warcraft grid paths using map images. The second assesses path prediction performance with varying graph sizes and number of excluded nodes in the graph sampling, using synthetic data. The third applies path prediction to actual taxi trajectories. We demonstrate our method's ability to learn from both optimal and suboptimal trajectories, achieving efficiency in scenarios with numerous trajectories for identical or similar contexts. We also provide examples of generated trajectories between nodes and possible destinations given a partial paths.

### 4.1 WARCRAFT MAPS EXPERIMENT

We replicated the experiment from Pogančić et al. [2020] to predict optimal paths on 18x18 Warcraft grids ($V = 324$) using 144x144 (Figure 3a) map images. The base experiment W aims to predict optimal paths from the upper left to the bottom right corner, similar to the original study. In W-M2M we input actual terrain weights (Figure 3c) to Dijkstra's algorithm to produce 30 optimal paths (displaying three in Figure 3b) per image from random nodes on upper/left edges to random nodes on the opposite edges, using the first 350 images for consistent training data volume. As last step W-M2M-N introduces noise into the terrain

| Method | Synthetic ($|V| = 30$) | | Synthetic ($|V| = 50$) | | Synthetic ($|V| = 100$) | | Cabspot ($|V| = 355$) | |
|---|---|---|---|---|---|---|---|---|
| | Jacc (%) | Match (%) | Jacc (%) | Match (%) | Jacc (%) | Match (%) | Jacc (%) | Match (%) |
| PRIOR | 44.3 (0.4) | 26.7 (0.2) | 41.9 (0.8) | 17.8 (0.9) | 32.5 (0.5) | 9.4 (0.4) | 26.5 (0.9) | 13.5 (0.7) |
| FCNN | 56.9 (0.2) | 27.1 (0.6) | 51.6 (0.7) | 14.0 (1.0) | 49.0 (0.3) | 5.0 (1.0) | 15.7 (0.5) | 10.7 (0.4) |
| DBCS | 66.4 (2.1) | 45.4 (3.3) | 57.2 (2.5) | 32.8 (2.1) | 34.2 (0.6) | 10.0 (0.8) | 38.5 (1.2) | 15.9 (2.6) |
| DataSP | **76.6 (0.8)** | **63.9 (1.2)** | **73.1 (0.6)** | **52.6 (1.0)** | **67.7 (0.3)** | **37.3 (2.0)** | **47.3 (1.2)** | **21.5 (1.0)** |

Table 2: Average of Jaccard Index (edges) and Match, i.e., % of optimal paths, are reported. Optimal paths based on learned costs are considered. Standard deviations are computed over five random starts in the graph and data generation.

weights ($y \leftarrow max(0, y(1 + 0.5\epsilon))$), generating 30 suboptimal paths per image for training. The objective across the three variants is to predict optimal paths for 1000 test images from upper left to bottom right as in the original experiment. Our approach involves graph sampling 100 out of 324 nodes per learning iteration, comparing it with the Differentiable Blackbox Combinatorial Solver (DBCS) proposed in Pogančić et al. [2020], and integrating the first five ResNet18 layers into DataSP (and DBCS), without assuming prior grid cost knowledge ($\mathcal{L}_P = 0$). Additional details and code are in Appendix D and supplementary material.

**Results: Optimal Paths** After the learning is complete, ResNet18 is able to reconstruct grid weights, closely matching the actual ones, as shown in Figure 3d compared to 3c. This allows any exact Shortest Path solver to predict optimal paths based on the inferred costs, as shown in Figure 4c. Table 1 contains the percentage of inferred trajectories (solved by Dijkstra on the learned edges' costs) having optimal path cost as done in the original experiment. We first show that our results are competitive to the baseline in a single source-target setting considering optimal observed paths. We then show that, while both methods have good performance considering many to many nodes, the time computation of our method does not increase with the number of observed trajectories per image, while the baseline increases linearly. Finally, our method shows to be considerably better on learning from suboptimal trajectories.

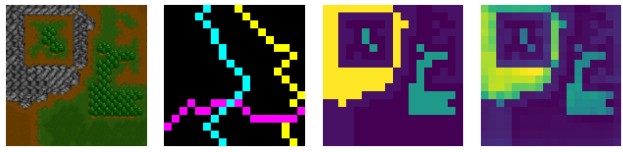

(a) Images as context features (b) Observed paths (c) True latent costs (d) Pred. latent costs

Figure 3: Algorithm 3 gets several images and paths as training data to reconstruct the grid costs with the trained Resnet18.

**Results: Path Distribution** By inputing the inferred $P^\omega$ to Algorithm 4 several times, we do Monte Carlo simulation to provide an approximation of the distribution of likely

paths from a given start to a given target node, as shown in Figure 4d. We are also able to provide likely destination nodes given a new context and a partial path, as shown in Figure 5. In this case, we provide three different priors of destination nodes. The first prior is a uniform distribution, assuming that all the nodes are equally likely to be a destination. In the second we assume distant nodes, in terms of predicted costs, are exponentially less likely than closer nodes. In the third, we assume there a uniform distribution through all nodes in the bottom of the grid. The full sequence of destination predictions is provided in Appendix E, Figure 9.

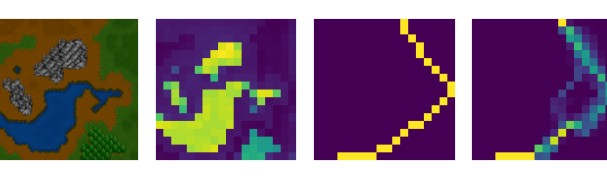

(a) Test image (b) Pred. costs (c) Pred. path (d) Likely paths

Figure 4: Algorithm 4 gets $P_n^\omega$ as input to provide likely paths between given source and target nodes, where $n$ represents the given image and $\omega$ the trained Resnet18 weights.

## 4.2 ROUTES EXPERIMENTS

In this subsection, we present both synthetic and real-dataset route experiments.

**Synthetic Dataset.** In our synthetic data generation process, we emulate real-world graph complexity. Initially, we construct a binary adjacency matrix based on node count and a sparsity-defining scalar, indicating node connectivity. We then simulate real-world edge cost complexities using a noisy, nonlinear function with conditional features from normal distributions, introducing multimodal noise to mirror cost uncertainties. Subsequently, we apply a biasing technique limiting source-target node pair permutations, reflecting real-world data patterns. Lastly, using Dijkstra's algorithm, we calculate the shortest path to represent each 'observed suboptimal path', considering the generated edge costs and node pairs. This approach allows us to control the graph and data properties. We generate three

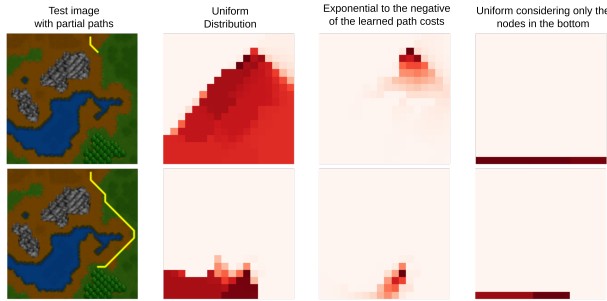

Figure 5: Given two partial paths and a map as a context in the test data, we compute the probability of each node to be a destination based on three different priors.

| Method | $|V_s|/|V|$ | $|V| = 50$ | $|V| = 100$ |
|---|---|---|---|
| PRIOR | - | 41.9 | 32.4 |
| DBCS | - | 57.2 | 34.2 |
| DataSP | 20% | 66.5 | 58.8 |
| DataSP | 30% | 67.5 | 62.3 |
| DataSP | 50% | 71.4 | 65.1 |
| DataSP | 100% | 73.1 | 67.7 |

Table 3: DataSP outperforms DBCS even when node sampling is reduced to 20% for memory efficiency.

datasets. The first represents a graph with $|V| = 30$ and $|E| \approx 270$, the second $|V| = 50$ and $|E| \approx 750$, both with 5000 training samples of features and respective paths for a sampled source-target nodes within a limited number of source-target combinations. The third set contains $|V| = 100$ and $|E| \approx 3000$, with 15000 training samples.

**Real Dataset** We used the Cabspotting dataset [Piorkowski et al., 2009]. Our analysis focused on San Francisco's central region, selected for its diverse range of source-target node combinations in the dataset's observed paths. To build the graph, we generated a grid over our defined region of interest. Each grid square was considered a candidate for a node if the number of observed taxi geolocations (determined by latitude and longitude data) exceeded a predetermined threshold. Edges between nodes were identified based on the presence of at least one trip traversing between the two nodes. Our graph ended up with 355 nodes and 2178 edges. To ensure comprehensive edge detection, taxi geolocations were linearly interpolated. We removed trajectories with cycles for simplicity.

**Baselines.** We evaluate three primary baselines. The first is the **PRIOR** framework, employing the shortest path algorithm with edge costs determined by Euclidean distances between connected nodes, denoted as $\boldsymbol{y}^p$. The second baseline is a fully connected neural network (**FCNN**) that directly predicts edge usage. Our third baseline is **DBCS**. We tuned the hyperparameter $\lambda$ of DBCS for our experiments. Additionally, we used their method for prior regularization, which gave better results in our experiments. We connect the same FCNN architecture to the DBCS and to DataSP, which is composed of three hidden layers of 1024 neurons each, with ReLU activation functions in the hidden layers. The choice of baselines is further discussed in Appendix D.1.

**Results: Expected Optimal Paths.** In Table 2, we compare the expected optimal path from DataSP to the predicted path of the baselines. The *Match* metric indicates the per-

centage of predicted paths matching exactly with the paired observed paths in the test data. The *Jacc* represents the average Jaccard Index of edges through the test dataset. DataSP outperforms the baselines and generalizes better for larger graphs. Similar to the findings in Pogančić et al. [2020], our work also highlights the limitations of traditional NN approaches in generalizing for complex and structured predictions, i.e., they cannot guarantee the edges are connected and lead from the source to end nodes. FCNN performs even worse than the considered prior for larger graphs as they are not able to guarantee path feasibility. Additional results of path distribution in the graph are illustrated in Appendix E.

**Results: Graph Sampling.** Table 3 reveals that while reducing node sampling in training lowers performance, our method still achieves a high Jaccard index compared to baseline algorithms. This trade-off enables scalability to larger graphs with minimal performance loss. For instance, in the 100-node experiment, reducing to 30 nodes per iteration drastically decreases space complexity to 2.7% of the original, with a performance drop from 0.677 to 0.623 but still way above our baseline.

## 5 CONCLUSION

We proposed DataSP, a differentiable and probabilistic algorithm to solve all to all shortest path problems. We have shown how to connect this algorithm with neural networks to allow learning from optimal or suboptimal trajectories. By learning from these trajectories we are able to infer latent costs of graph connections. We demonstrated that these costs can be used to either plan an optimal path, or to predict path distributions with DataSP. We have proven that smoothing the max and argmax operator in the DataSP leads the path distribution behavior to be aligned to the maximum entropy principle. We have also shown that graph sampling in the learning process can be extremely beneficial to balance the trade-off between performance and computational resources. Finally, we outline that DataSP is very efficient to learn from datasets with big number of trajectories due to its access to all start and target combinations in a single run.

**Acknowledgements**

This work has been supported by the Industrial Graduate School Collaborative AI & Robotics funded by the Swedish Knowledge Foundation Dnr:20190128, and the Knut and Alice Wallenberg Foundation through Wallenberg AI, Autonomous Systems and Software Program (WASP).

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

# DataSP: A Differential All-to-All Shortest Path Algorithm for Learning Costs and Predicting Paths with Context (Supplementary Material)

Alan A. Lahoud[1]     Erik Schaffernicht[1]     Johannes A. Stork[1]

[1]Center for Applied Autonomous Sensor Systems (AASS), Örebro University, Örebro, Sweden

## A  EFFICIENT DATASP

**Minimizing Redundant Computations.**  We incorporate the following simplifications in DataSP to avoid tracking unnecessary operations for backpropagation with autograd:

- Self-Loop Elimination: When $i = j$, the cost matrix remains unchanged, as our model presupposes non-negative edge costs. This is ensured by constraining the learned latent variables in $M$, representing edge costs, to be strictly positive.

- Direct Path Redundancy: In cases where $i = k$ or $k = j$, the algorithm evaluates direct paths between nodes $i$ and $j$. Since these direct paths are already set in the initialization of $M$ and $P$, we bypass updates.

- Infinite Path Pruning: If either $M[i, k]$ or $M[k, j]$ holds an infinite value, it indicates the absence of a viable shortcut path. We then bypass updates as any potential shortcut would not offer a shorter route than the direct path.

**Vectorizing DataSP.**  We also vectorize the inner loops $(i, j)$ leading to a significant reduction in computational time, as shown in Algorithm 5. This vectorization is feasible because there is no interdependence between the loop values within each $k$ iteration as we explain below:

- The update of $M[i, j]$ depends on the current values of $M[i, k]$, $M[k, j]$, and $M[i, j]$ itself. The value $M[i, j]$ is not yet updated in the $k^{th}$ iteration, so there is no interdependency here.

- $M[i, k]$ was already evaluated for update if $k < j$ and the same for $M[k, j]$ if $k < i$. However, even on these cases, no actual update occurs because $M[i, k]$ relies on the previous value of $M[i, k]$ itself and $M[k, k]$ only, which is guaranteed of not having an update based on our avoidance of redundant computations detailed above. This is analogous for computing $M[k, j]$.

---

**Algorithm 5** Efficient DataSP

---

**Input:** A cost matrix $M \in \mathbb{R}_{+}^{|V_S| \times |V_S|}$
**Output:** Shortcut Probabilities $P \in [0, 1]^{|V_S| \times |V_S| \times |V_S|}$

1: Init $P[i, j, k] = 1$ if $i = k$ and $M[i, j] < inf$, $0$ otherwise
2: **for** $k = 0$ to $|V_S| - 1$ $\forall i, j$ without redundancy, in parallel **do**
3:     $(p_s, p_d) \leftarrow \Phi_\beta(M[i, k] + M[k, j], M[i, j])$
4:     $P[i, j, k] \leftarrow p_s$
5:     $P[i, j, c] \leftarrow p_d P[i, j, c]$   $\forall c < k$ and $c = i$
6:     $M[i, j] \leftarrow \min_\beta(M[i, k] + M[k, j], M[i, j])$
7: **end for**
8: **return** $P$

---

# B  PROOFS

## B.1  PROOF OF THEOREM 1

From the vectorized DataSP algorithm, given an initial $M^0$ cost matrix, we compute $M^1[i,j]$ in the first iteration as

$$M^1[i,j] = \min_\beta(M^0[i,1] + M^0[1,j], M^0[i,j]) \tag{5}$$

The first argument of $\min_\beta$ is the cost of going from $i$ to $j$ through 1, while the second is the cost of going from $i$ to $j$ without passing through 1, considering only the node 1 as a possible intermediate node. This can be rewritten as

$$M^1[i,j] = \min_\beta(\boldsymbol{f^1}_{i\to j}) \tag{6}$$

where $\boldsymbol{f^1}_{i\to j}$ is a vector containing the costs of all possible costs from $i$ to $j$ considering 1 as the only intermediate node.

In the second iteration of DataSP, we have

$$
\begin{aligned}
M^2[i,j] &= \min_\beta(M^1[i,2] + M^1[2,j], M^1[i,j]) \\
&= \min_\beta(\min_\beta(\boldsymbol{f^1}_{i\to 2}) + \min_\beta(\boldsymbol{f^1}_{2\to j}), \min_\beta(\boldsymbol{f^1}_{i\to j})) \\
&= \min_\beta(M^0[i,1] + M^0[1,2] + M^0[2,1] + M^0[1,j], \\
&\qquad\quad M^0[i,1] + M^0[1,2] + M^0[2,j], \\
&\qquad\quad M^0[i,2] + M^0[2,1] + M^0[1,j], \\
&\qquad\quad M^0[i,1] + M^0[1,j], \\
&\qquad\quad M^0[i,0] + M^0[0,j], \\
&\qquad\quad M^0[i,j]) \\
&= \min_\beta(\boldsymbol{f^2}_{i\to j})
\end{aligned}
\tag{7}
$$

where $\boldsymbol{f^2}_{i\to j}$ is a vector containing the costs of all possible costs from $i$ to $j$ considering 1 and 2 as the only intermediate nodes.

For a general $k$,

$$
\begin{aligned}
M^k[i,j] &= \min_\beta(M^{k-1}[i,k] + M^{k-1}[k,j], M^{k-1}[i,j]) \\
&= \min_\beta(\min_\beta(\boldsymbol{f^{k-1}}_{i\to k}) + \min_\beta(\boldsymbol{f^{k-1}_{k\to j}}), \min_\beta(\boldsymbol{f^{k-1}_{i\to j}})) \\
&= \min_\beta(\boldsymbol{f^k_{i\to j}})
\end{aligned}
\tag{8}
$$

Note that $M^2[i,j]$ was calculated using the $\log e^x = x$ and $\log(ab) = \log a + \log b$, which is analogous for any $k$.

**Bias in visited cycles.**    The resulting paths contains all paths going from $i$ to $k$ without cycles plus paths from $k$ to $j$ without cycles. However, the concatenation of these paths can result in paths containing cycles, e.g., $i \to k-1 \to k \to k-1 \to j$. In practice, we remove cycles to avoid bias, e.g., while the path $i \to k-1 \to k \to k-1 \to j$ is visited, $i \to k \to k-1 \to k \to j$ is not.

## B.2  PROOF OF THEOREM 2

In the $k^{th}$ iteration of the vectorized DataSP, $P^k[i,j,k]$ is computed as $\Phi(-M^{k-1}[i,k] - M^{k-1}[k,j], -M^{k-1}[i,j])$. From 1, we can rewrite it as $\Phi(-\boldsymbol{f^{k-1}_{i\to k}} - \boldsymbol{f^{k-1}_{k\to j}}, -\boldsymbol{f^{k-1}_{i\to j}})$. Using logarithm and exponential properties, we rewrite it as

$$P^k[i,j,k] = \frac{\sum e^{-\boldsymbol{f^{k-1}_{i\to k}}} \sum e^{-\boldsymbol{f^{k-1}_{k\to j}}}}{\sum e^{-\boldsymbol{f^{k-1}_{i\to k}}} \sum e^{-\boldsymbol{f^{k-1}_{k\to j}}} + \sum e^{-\boldsymbol{f^{k-1}_{i\to j}}}} \tag{9}$$

In the numerator, applying the distributive multiplication property, the resulting value is the sum of the exponential of all possible paths going from $i$ to $j$ passing through $k$ considering a subgraph of nodes from 1 to $k-1$ in each subpath. Since we avoid visiting paths with cycles, we can consider also the node $k$ in the subgraph to simplify the formula. We then rewrite the above equation as

$$P^k[i,j,k] = \frac{\sum e^{-f^k_{i\to k\to j}}}{\sum e^{-f^k_{i\to k\to j}} + \sum e^{-f^{k-1}_{i\to j}}} \tag{10}$$

where $f^k_{i\to k\to j}$ is a vector containing the costs of all paths from $i$ to $j$ passing through $k$, computed considering only the subgraph of nodes from 1 to $k$.

Now, while the first term of the denominator contains all paths from $i$ to $j$ going through $k$, the second term of the denominator contains all paths from $i$ to $j$ that does not go through $k$, because it is computed considering the subgraph of nodes 1 to $k-1$. Thus, the resulting term should contain all the paths from $i$ to $j$ considering the subgraph 1 to $k$, and we rewrite the equation above as

$$P^k[i,j,k] = \frac{\sum e^{-f^k_{i\to k\to j}}}{\sum e^{-f^k_{i\to j}}} \tag{11}$$

Analogously, we can derive the probability that $k$ is not an intermediate node between $i$ and $j$ considering the same subgraph, which results in

$$P^k[i,j,\neg k] = \frac{\sum e^{-f^{k-1}_{i\to j}}}{\sum e^{-f^k_{i\to j}}} \tag{12}$$

To further analyze the formula for a general $k'$, we first compute the probability that $k-1$ is the highest intermediate node in the $k^{th}$ iteration, which is the product of the probability that $k-1$ is the highest intermediate node in the $(k-1)^{th}$ iteration and the probability that $k$ is not the highest intermediate node in the $k^{th}$ iteration:

$$P^k[i,j,k-1] = P^{k-1}[i,j,k-1]P^k[i,j,\neg k]$$
$$= \frac{\sum e^{-f^{k-1}_{i\to k-1\to j}}}{\sum e^{-f^{k-1}_{i\to j}}} \frac{\sum e^{-f^{k-1}_{i\to j}}}{\sum e^{-f^k_{i\to j}}} = \frac{\sum e^{-f^{k-1}_{i\to k-1\to j}}}{\sum e^{-f^k_{i\to j}}} \tag{13}$$

For the node $k' = k - C$, considering $C < k$, we have

$$P^k[i,j,k'] = P^{k'}[i,j,k'] \prod_{c=1}^{C-1} P^{k-c}[i,j,\neg(k-c)] \tag{14}$$

The terms are simplified and we finally have

$$P^k[i,j,k'] = \frac{\sum e^{-f^{k'}_{i\to k'\to j}}}{\sum e^{-f^k_{i\to j}}}. \tag{15}$$

The proof above omitted $\beta$ without loss of generality, as we could simply replace $f$ to $\beta f$.

## B.3 PROOF OF THEOREM 3

Given a graph with $|V|$ nodes, it is direct from theorem 2 that the probability to sample the node $H$ from $i$ to $j$ in Algorithm 4 is $\frac{\sum e^{-f^H_{i\to H\to j}}}{\sum e^{-f^{|V|}_{i\to j}}}$. If $H = i$, the sampling process ends indicating a direct path from $i$ to $j$. Otherwise, the sampling process

continues recursively. The next sampling iteration are splint into two subpaths: between $i$ and $H$, and between $H$ and $j$. Again, the probability to sample the node $H_L$ from $i$ to $H$ is $\frac{\sum e^{-f_{i \to H_L \to H}^{H_L}}}{\sum e^{-f_{i \to H}^{H}}}$. Analogously, the probability to sample the node $H_R$ from $H$ to $j$ is $\frac{\sum e^{-f_{H \to H_R \to j}^{H_R}}}{\sum e^{-f_{H \to j}^{H}}}$. After this iteration, the probability to have a sampled path $i \to H_L \to H \to H_R \to j$ is the product of the probabilities above:

$$\frac{\sum e^{-f_{i \to H \to j}^{H}}}{\sum e^{-f_{i \to j}^{|V|}}} \frac{\sum e^{-f_{i \to H_L \to H}^{H_L}}}{\sum e^{-f_{i \to H}^{H}}} \frac{\sum e^{-f_{H \to H_R \to j}^{H_R}}}{\sum e^{-f_{H \to j}^{H}}}. \tag{16}$$

Using logarithm properties, we conclude that $\sum e^{-f_{i \to H}^{H}} \sum e^{-f_{H \to j}^{H}} = \sum e^{-f_{i \to H \to j}^{H}}$, simplifying the equation above to

$$\frac{\sum e^{-f_{i \to H_L \to H}^{H_L}} \sum e^{-f_{H \to H_R \to j}^{H_R}}}{\sum e^{-f_{i \to j}^{|V|}}} \tag{17}$$

After splitting the path into $c$ subpaths, we have

$$\frac{\prod_{c=1}^{C} \sum e^{-f_{N_{c-1} \to N_{Hc} \to N_c}^{N_c}}}{\sum e^{-f_{i \to j}^{|V|}}} \tag{18}$$

where $N_0 = i$ and $N_C = j$.

This process is done until $N_{Hc} = N_{c-1}$ for all $c$ in the path, representing a direct subpath, indicating the end of the sampling process in the branch. The direct edges cost can be denoted by the initial cost matrix $M$ of the DataSP algorithm. Replacing the vector $-f_{N_{c-1} \to N_c \to N_{c+1}}^{N_c}$ with $-M[c-1, c+1]$, the sum in the numerator is no longer useful. Leveraging the property that the product of exponentials are the exponentials of the sum, we can finally write the probability to sample a path $N_0 \to N_1 \to ... \to N_C$ as

$$\frac{e^{-\sum_{c=1}^{C} M[c-1,c]}}{\sum e^{-f_{i \to j}^{|V|}}}. \tag{19}$$

The proof above omitted $\beta$ without loss of generality, as we could simply replace $f$ to $\beta f$.

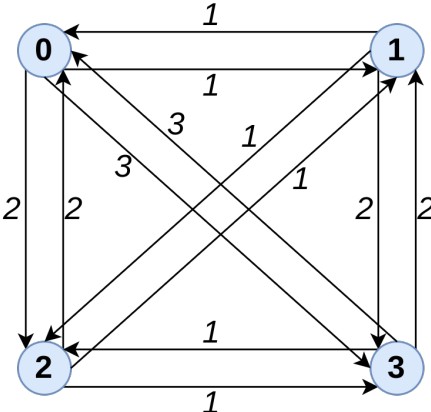

Figure 6: Illustrative example of graph to evaluate the space visited paths in DataSP.

## C  ILLUSTRATIVE EXAMPLE OF PATH SAMPLING

**Illustrative graph.**    We run a simple demonstration on the graph of Figure 6 to show the frequency of the sampled paths to go from node 0 to node 3. The edges' cost are set as $M_{ij} = |i - j|$ for all nodes $i \neq j$ and $\infty$ if $i = j$. We first run DataSP with the described $M$ and $\beta = 1$ to obtain $P$. Then, we run Algorithm 4 10000 times.

**Paths frequency.**    The frequency of paths sampled from it is described in Table 4. The frequency is aligned with the theoretical probability derived in Theorem 3.

**Intuition of non-visited cycles.**    Consider $H[i, j]$ the *highest intermediate node* between $i$ and $j$. Note that the walk $0 \rightarrow 2 \rightarrow 0 \rightarrow 2 \rightarrow 3$ has cost 7 and is not a visited cycle (not shown in Table 4). If we try to reach this walk in the sampling, we first need to set $H[0, 3] = 2$ leading it to $0 \rightarrow 2 \rightarrow 3$. We can only expand the left part with either $H[0, 2] = 1$ or $H[0, 2] = 0$. The first does not match with our desired walk while the second terminates the recursion in Algorithm 4. We can only expand the right part with $H[2, 3] = 0$, $H[2, 3] = 1$, or $H[2, 3] = 2$. $H[2, 3] = 2$ terminates the recursion while $H[2, 3] = 1$ does not match with our desired walk. By choosing $H[2, 3] = 0$ we can't have any other additional node between 2 and 3 since 0 would be the highest one, so it leads to the end of the recursion as well without reaching the desired path. In practice, we do not consider walks containing cycles. In Table 4, for instance, the only desired paths to sample is the first four paths.

## D  IMPLEMENTATION DETAILS

NNs were trained on Nvidia GeForce RTX 4070 GPU. We used Pytorch with Adam optimizer in all implementations. To have approximately the same results as in the paper, the hyperparameters in Table 5 should be followed. To further reproduce the numbers one should choose the "seed" input values from 0 to 5 together with those hyperparameters, selecting the best model according to validation set results, which were then reported on the test set.

For the Warcraft experiments, we did not use any prior knowledge for the latent costs, while in the Route experiments, we used. In the Cabspotting dataset, we used the euclidean distance between nodes to be the $y^p$. For the methods with $y^p \neq 0$, we set the neural network to learn the difference between $y^p$ and the latent cost. We have also used the prior in the DBCS for a fair comparison, as ir resulted in a better performance than not using it.

For the baseline DBCS, we had to iteratively compute the shortest path between different start and end nodes, increasing the time computation even when having various trajectories for each context.

For the route experiments, each trajectory was connected to a single contextual feature. To aggregate $c$ trajectories to each context, we aggregated paths for similar contexts. For example, trajectories happened in Saturday at 10:00 and Saturday at 10:05 would be probably set together in the same batch of trajectories. More specifically, for each context sample feature, we select the top 1% most similar $x$ samples to a particular $x_n$, and then we pick their respective trajectories to be encoded by the same batch through $F$. We computed the similarity by calculating the euclidean distance between features in continuous

| $\tau_{0 \to 3}$ | Cost | Freq. | $\frac{exp(\text{-Cost})}{Z}$ |
|---|---|---|---|
| $0 \to 3$ | 3 | .2153 | .2136 |
| $0 \to 1 \to 3$ | 3 | .2141 | .2136 |
| $0 \to 2 \to 3$ | 3 | .2119 | .2136 |
| $0 \to 1 \to 2 \to 3$ | 3 | .2097 | .2136 |
| $0 \to 1 \to 0 \to 2 \to 3$ | 5 | .0310 | .0289 |
| $0 \to 1 \to 2 \to 1 \to 3$ | 5 | .0298 | .0289 |
| $0 \to 2 \to 1 \to 3$ | 5 | .0293 | .0289 |
| $0 \to 1 \to 0 \to 3$ | 5 | .0288 | .0289 |
| $0 \to 1 \to 2 \to 0 \to 1 \to 3$ | 7 | .0050 | .0039 |
| $0 \to 1 \to 0 \to 2 \to 1 \to 3$ | 7 | .0050 | .0039 |
| $0 \to 2 \to 0 \to 3$ | 7 | .0042 | .0039 |
| $0 \to 1 \to 2 \to 1 \to 0 \to 3$ | 7 | .0038 | .0039 |
| $0 \to 2 \to 1 \to 0 \to 3$ | 7 | .0035 | .0039 |
| $0 \to 2 \to 0 \to 1 \to 3$ | 7 | .0032 | .0039 |
| $0 \to 1 \to 2 \to 0 \to 3$ | 7 | .0031 | .0039 |
| $0 \to 1 \to 0 \to 2 \to 0 \to 1 \to 3$ | 9 | .0010 | .0005 |
| $0 \to 2 \to 0 \to 1 \to 0 \to 3$ | 9 | .0005 | .0005 |
| $0 \to 1 \to 2 \to 0 \to 1 \to 0 \to 3$ | 9 | .0003 | .0005 |
| $0 \to 1 \to 0 \to 2 \to 0 \to 3$ | 9 | .0003 | .0005 |
| $0 \to 1 \to 0 \to 2 \to 1 \to 0 \to 3$ | 9 | .0002 | .0005 |

Table 4: Comparison between the Monte Carlo simulation through DataSP to approximate paths' distribution and the theoretical values coming from the maximum entropy principle.

| Hyperparams/Exp | W | W-M2M | W-M2M-N | Synthetic | Cabspotting |
|---|---|---|---|---|---|
| Learning Rate | 0.0002 | 0.001 | 0.001 | 0.0001 | 0.0001 |
| $\beta$ | 100 | 100 | 30 | 1 | 30 |
| Batch size | 24 | 16 | 8 | 16 | 32 |
| $|V_s|/|V|$ | 70/324 | 100/324 | 100/324 | Varied | 70/355 |
| $c$ | 1 | 30 | 30 | 1% Data | 1% Data |
| $\alpha$ | 0 | 0 | 0 | $10^{-5}$ | $10^{-5}$ |

Table 5: Hyperparameters used to report results in the paper.

space and hamming distance for discrete features.

For the graph sampling, instead of selecting the nodes totally randomly, we select half nodes to be connected to each other (connected subgraph), and the other half randomly based on its frequency of appearance in the training data. Here, the heuristics was arbitrary. The performance using this strategy was slightly better than choosing totally random nodes. Note that by sampling nodes, we have to make sure that paths called during the training data go through at least 2 of these nodes, which is done by simply filtering paths using this rule.

## D.1 CHOICE OF BASELINES

We opted to choose a differentiable combinatorial blackbox (DBCS) from Pogančić et al. [2020] as a baseline, that has demonstrated to be efficient to approximately differentiate through the Dijkstra algorithm, as it is an efficient algorithm to solve the shortest path problem. This baseline allows us to capture the limitation of other works, which is not focusing on multiple source and target nodes.

As discussed with the reviewers, we have also explored other baselines without success in our experiments. A differentiable linear programming version of the shortest path led to significant convergence issues, preventing us from presenting those results. Our efforts here were based on the same linear programming formulation used in Cristian et al. [2023] with differentiable cvxpy layers.

Another potential baseline would be the perturbed Fenchel-Young loss (PFYL) proposed in [Berthet et al., 2020]. However, their implementation is already quite time-consuming on a 12x12 grid for a single source-target scenario, as evidenced by running the collection of differentiable algorithms from Tang and Khalil [2022]. This time consumption likely arises from the perturbation sampling process during the training procedure.

# E ADDITIONAL RESULTS

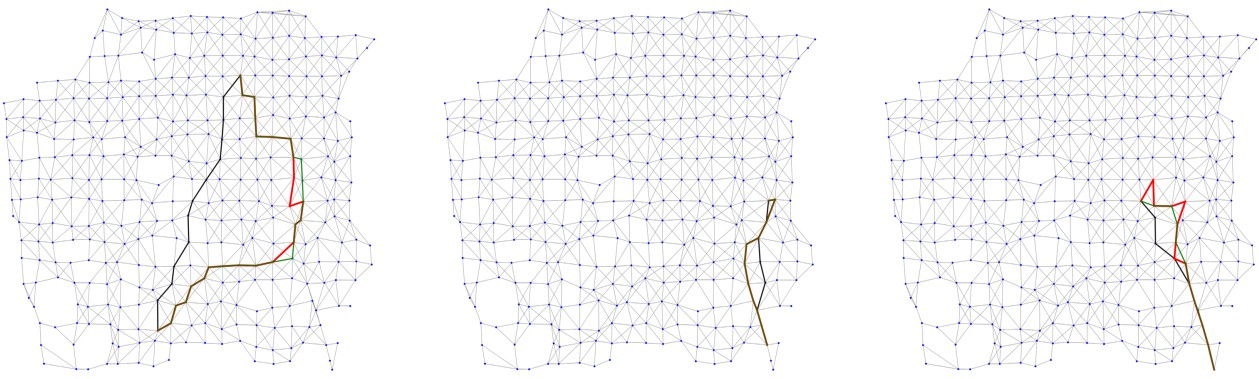

Figure 7: Examples of path predictions in Cabspotting. Green paths represent the most likely path prediction from DataSP given test contextual features. Red are observed paths for the same contextual features. Black are paths computed from PRIOR. Edges orientation were omitted for better visualization.

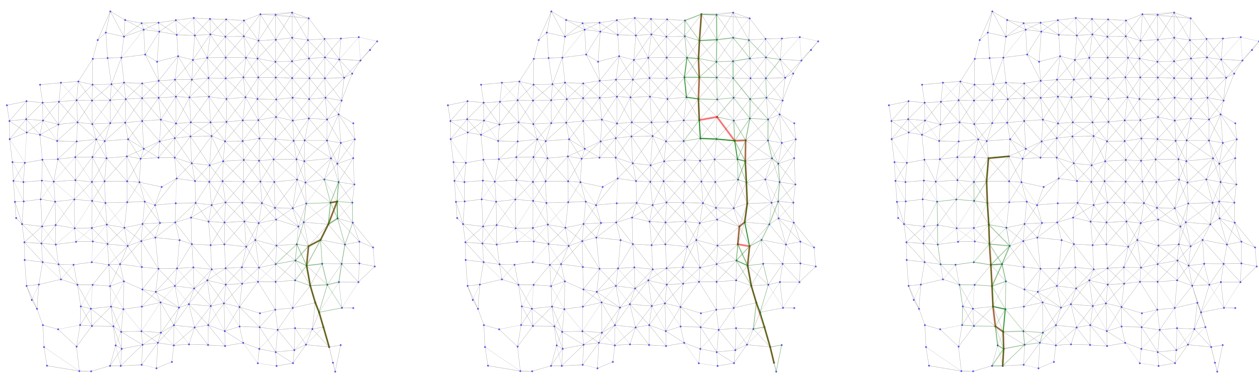

Figure 8: Examples of path distribution predictions. Green paths represent predicted distribution from DataSP given test contextual features, the darker the green, the more likely is the prediction. Red are observed paths for the same contextual features. Edges orientation were omitted for better visualization.

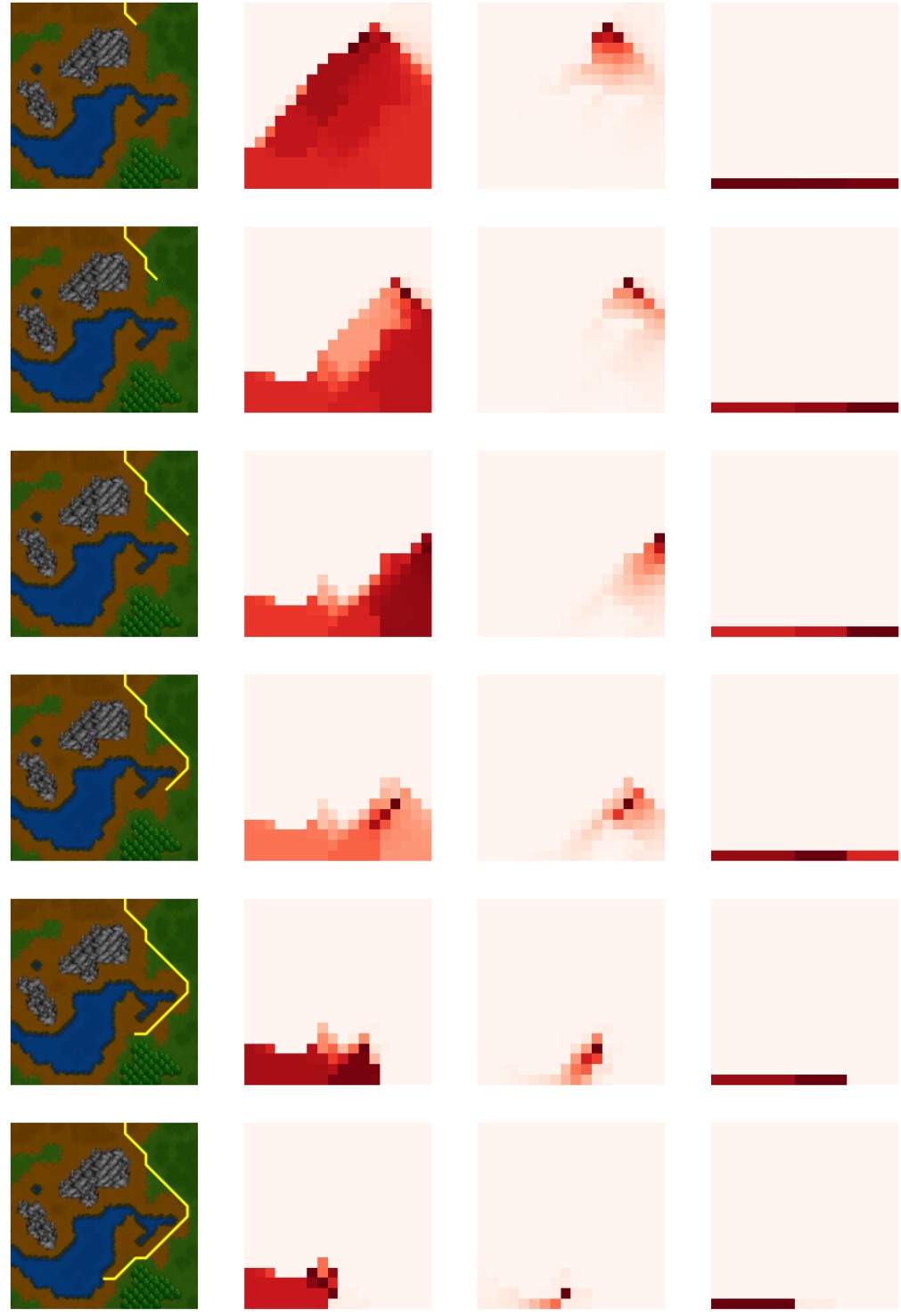

Figure 9: A sequence of simulations to compute likely destinations given a partial path (first column). Second, third, and fourth columns represent different priors on destination likelihood. Second column with uniform, third with expneg, and fourth with uniform only in the bottom grid row.