# OpenReview forum: "DataSP: A Differential All-to-All Shortest Path Algorithm for Learning Costs and Predicting Paths with Context"
_auai.org/UAI/2024/Conference — UAI 2024 poster_

### Official Review · Reviewer_6Jei · 2024-03-25

**Q2-1 Originality-Novelty:** 2
**Q2-2 Correctness-Technical Quality:** 3
**Q2-5 Clarity Of Writing:** 3

**Q1 Summary And Contributions:**

This paper introduces DataSP, an approach for learning latent costs of transitions on graphs from trajectory demonstrations with contextual features. DataSP is a differentiable all-to-all shortest path algorithm that efficiently handles a large number of trajectories in each learning step. It accommodates complex latent cost functions through neural network approximation. The paper also presents a path sampling method from DataSP, demonstrating that the inferred distribution follows the maximum entropy principle. Experimental results show that DataSP outperforms state-of-the-art methods in predicting paths on graphs, highlighting its effectiveness for path-planning tasks

**Q2-3 Extent To Which Claims Are Supported By Evidence:**

2: Fair: the main claims are somewhat supported by evidence (but the experimental evaluation may be weak, or does not match entirely with the claims, important baselines may be missing, proofs contain important ideas but lack rigor, algorithmic details are only discussed superficially, references are imprecise, assumptions are not sufficiently motivated or explicated, etc.).

**Q2-4 Reproducibility:**

2: Fair: key resources (e.g. proofs, code, data) are unavailable but key details (e.g. proof sketches, experimental setup) are sufficiently well-described for an expert to confidently reproduce the main results.

**Q3 Main Strengths:**

The paper excels in its clear and accessible writing style, ensuring readers can easily comprehend and follow its content. Moreover, the problem tackled in the paper is considered somewhat intriguing and holds significance within its domain.

The exploration of making the Floyd-Warshall Algorithm differentiable through techniques such as smooth min shows promise.

**Q4 Main Weakness:**

The novelty of this paper may be somewhat constrained. From my standpoint, the primary contribution lies in rendering the Floyd-Warshall Algorithm differentiable. Apart from this aspect, I find the remaining ideas to be intuitive and conventional.

**Q5 Detailed Comments To The Authors:**

See my above comments.

**Q9 Complying With Reviewing Instructions:**

Yes

---

> ### Author Rebuttal · Authors · 2024-04-03
>
> We are grateful for the reviewer's insightful comments and the opportunity to discuss the contributions of our paper further.
>
> We acknowledge that the main contribution is making the FW algorithm differentiable. However, we not only make FW differentiable by implementing the smooth min and argmin, but we also prove that by doing this, the FW is transformed to its probabilistic version, capable to sample paths (through Algorithm 4) that are aligned to the Maximum Entropy Principle (Theorem 3). The derivations of Theorem 3 is not trivial, as it depends on the biased order of the FW loops, and therefore we needed to carefully define the inferred ($P$) and observed ($F$) frequency of the “highest intermediate node”. The proof of Theorem 3 is a consequence of Theorems 1 and 2, meaning that the learning phase is totally connected to the sampling algorithm, and therefore showing that DataSP is useful both for learning and probabilistic inference.
>
> We also believe that the graph sampling part (Section 3.3) is an important aspect of the contribution.
> While graph sampling itself is not a new concept, our work innovates by introducing a method to simplify the graph without compromising the "smooth" property during the learning phase. This ensures that the inferred “smooth optimal” costs $M$, for the remaining edges, remain consistent as if no nodes were removed.

---

### Official Review · Reviewer_HBSE · 2024-03-26

**Q2-1 Originality-Novelty:** 3
**Q2-2 Correctness-Technical Quality:** 3
**Q2-5 Clarity Of Writing:** 4

**Q1 Summary And Contributions:**

This paper studies the inverse problem (not exactly but similar to inverse reinforcement learning) to recover unknown parameters in an optimization problem from demonstrated near-optimal solutions. The training data yd-is composed of a feature x, and a set of trajectories $\tau$ and potentially with the ground truth parameters $y$. The example studied in the paper is the shortest path problem, where a feature is associated to each problem and a set of near-optimal paths are given to recover the mapping of features to edge weights.

The paper follows the differentiable optimization approach to differentiate through the optimal solutions and back-propagate gradients. The main contribution is its model of near-optimal solutions. The paper assumes the shortest path solutions are formed in a softmax way, where the paths are selected with probability proportional to its exponential value using softmax. This model leads to the proposed modified Floyd-Warshall algorithm that modifies all the "max" operation to "softmax" with the same complexity $||V||^3.

Using this modified Floyd-Warshall algorithm (DataSP), the algorihtm becomes differentiable due to the use of softmax instead of max. The differentiability therefore allows the authors to end-to-end compare the learned parameters (edge weights) and their induced near-optimal solutions with the observed trajectories. The authors compare the distribution of the induced near-optimal solutions and the observed samples using KL divergence to define the loss function. Using this loss function and the differentiability of the FW algorithm, the authors end-to-end backprop gradients from the KL divergence loss through the algorithm and train the predictive model on predicting the edge weights. This end-to-end algorithm can directly learn a predictive model to produce predictions that match the observed samples. The authors also briefly discuss the computation challenge and a solution to address the challenge.

**Q2-3 Extent To Which Claims Are Supported By Evidence:**

3: Good: the main claims are supported by convincing evidence (in the form of adequate experimental evaluation, proofs, (pseudo-)code, references, assumptions).

**Q2-4 Reproducibility:**

3: Good: key resources (e.g. proofs, code, data) are available and key details (e.g. proofs, experimental setup) are sufficiently well-described for competent researchers to confidently reproduce the main results.

**Q3 Main Strengths:**

- Very clear presentation and theorem statements. It is indeed very well-written and consistent.
- To my knowledge, differentiating through FW algorithm, although simply based on using softmax to achieve differentiability, is new. There are many similar works using the same idea in the combinatorial optimization context though. For example, [1] uses softmax to make Frank-Wolfe algorithm differentiable. However, I think in the shortest path and the Floyd-Warshall algorithm context, it is a new contribution.

[1] "Differentiable Frank-Wolfe Optimization Layer", 21 Aug 2023  ·  Zixuan Liu, Liu Liu, Xueqian Wang, Peilin Zhao

**Q4 Main Weakness:**

- Baselines are not very convincing and unclear whether those are fair comparison. For example, "Prior" algorithm primarily uses $||y_n^\omega - y^P||$ to train the model and produce the optimal solution using the learned model and predicted edge weights. Given the assumption of the "near-optimal" demonstrations, a more fair comparison in the experiment is to use the learned model and the predicted edge weights to generate near-optimal shortest paths by running DataSP algorithm once (or any other similar algorithm). It looks like in the experiment (it is also not clearly specified so I can't tell), the "Prior" algorithm only uses the learned parameters to produce optimal instead of near-optimal solutions. This ignores the prior knowledge of the demonstrated trajectories are near-optimal only. Note that all these comparisons are in the test time only. So it doesn't require DataSP algorithm to be differentiable either. Similarly, "FCNN" and "DBCS" likely also only use its optimal solutions to compare with the near-optimal demonstrations, which also ignore the near-optimality knowledge. I didn't check the implementation but please explain and clarify your experiment details.

- Most works in the same type of problem in the same Warcraft application and in different applications indicate that "Prior" method usually works pretty well in simple predictive problem, where the Warcraft predictive problem is very simple given that the cost can be easily predicted based on the pixel color. Related to the previous question, could you explain why "Prior" performs so poorly in your case? Is it because of it ignores the "near-optimal" assumption?

- In addition to the above question about the baseline, there are also many MILP-based differentiable optimization end-to-end learning approach, e.g., [2]. These are newer than DBCS and also applicable to your setting. Could you explain why you didn't consider these MILP-based approach?

- The computation cost $||V||^3$, although with the speed up using subgraph sampling, is still quite expensive compared to MILP-based approach.

[2] Tang, Bo, and Elias B. Khalil. "Pyepo: A pytorch-based end-to-end predict-then-optimize library for linear and integer programming." arXiv preprint arXiv:2206.14234 (2022).

**Q5 Detailed Comments To The Authors:**

Please respond to the above weaknesses. I am happy to re-adjust my assessment given the authors' responses.

**Q9 Complying With Reviewing Instructions:**

Yes

---

> ### Author Rebuttal · Authors · 2024-04-03
>
> We thank the reviewer for their insightful feedback, allowing us to address potential misunderstandings, especially regarding the two first “main weaknesses” points.
>
> A. Clarification on $y^P$:
>
> Maybe there is some confusion about $y^P$ meaning. It does not represent training data for edge costs. Instead, we introduce $y^P$ as a constant vector representing the "prior knowledge" about edge values, as outlined in Section 1, "Minimization Problem". This concept is generally used in traditional inverse shortest path literature, often referred to as "initial arc costs" [1]. For our experiments, $y^P$ is assumed to be the Euclidean distances between nodes. We consider the inclusion of $y^P$ a reasonable assumption in real-world scenarios—e.g.,, knowing the Euclidean distances between landmarks. Thus, $y^P$ serves as an “initial guess” of edge costs, represented by a constant vector that doesn't change with the context. Note that, in order to preserve the integrity of the original Warcraft experiment design by [2], the Warcraft experiment does not utilize $y^P$.
>
> Where $y^P$ is assumed available (Cabspot and Synthetic experiments), it serves two roles. i) regularizing the main loss function in the learning algorithm through the term $L_p$, which we also used in DBCS for a fair comparison; ii) as an input to the PRIOR baseline (as described in the experiments). The PRIOR baseline is not a learning method, we simply input $y^P$ to a shortest path solver.
>
>
>
> B. Response to the specific points:
>
> Point 1 (Near-optimal):
>
> In Table 2, all the results considered the predicted “optimal” path in order to provide a fair comparison. This means we have learned the weights based on context and near-optimal paths (either using DataSP or baselines), and then chosen the most likely path (optimal) in the inference. In other words, although all the methods learn from near-optimal paths, in the test time only the optimal path is evaluated and compared.
>
>
> Point 2 (“PRIOR” baseline):
>
> For the "PRIOR" baseline, we didn't use a learning algorithm but directly inputted $y^P$ into a path solver. Here, our aim was to evaluate how much the learning methods (DataSP, DBCS and FCNN) performed better than an “initial reasonable guess”. The reason why the “PRIOR” baseline was not good was because the Euclidean distances between nodes, although somehow informative, don't fully match the underlying edges’ cost considered by the agents' when performing paths.
>
> Regarding the mentioned “simple predictive task”, this is exactly what we do with the “FCNN” baseline, i.e., predicting from context direct to path. This means predicting combinatorial solutions (or near-optimal) or structured predictions, which is not an easy task for traditional NN. This is observed in many important works such as [2] and [3] for the same Warcraft experiment. We confirmed that it leads to bad results with a Convolutional NN from images directly to paths, but we preferred not to report these results to prevent information overload. But we show that the FCNN (a direct prediction method) performs bad in the other experiments (Table 2).
>
>
>
> Points 3 and 4 (MILP):
>
> We appreciate the suggestion of MILP differentiable layers. Although unfamiliar with the specific paper mentioned, we did explore a linear programming (LP) version of the Shortest Path but faced significant convergence issues, which prevented us from presenting those results. Our efforts were based on the LP formulation in [4] and using cvxpy layers. Note that a direct comparison with [4] is not possible due to our reliance on demonstrations rather than training data on edge costs (grid costs in this case).
>
> Also, we limited our exploration of Differentiable MILP methods because they address problems with a single source and target node. Our work focuses on scenarios with multiple sources and targets, making these methods less applicable. We opted to use the adapted DBCS (to run multiple times and fit to “multiple sources and targets”) as our baseline because it incorporates the Dijkstra Algorithm, which is known to be specifically used for Shortest Path Problems.
>
>
>
> [1] Burton, Didier, and Ph L. Toint. "On an instance of the inverse shortest paths problem." Mathematical programming 53 (1992).
>
> [2] Vlastelica, Marin, et al. "Differentiation of blackbox combinatorial solvers." arXiv preprint arXiv:1912.02175 (2019).
>
> [3] Berthet, Quentin, et al. "Learning with differentiable pertubed optimizers." Advances in neural information processing systems 33 (2020).
>
> [4] Cristian, Rares, et al. "End-to-end learning for optimization via constraint-enforcing approximators." Proceedings of the AAAI Conference on Artificial Intelligence. 2023.

---

### Official Review · Reviewer_hHpY · 2024-03-26

**Q2-1 Originality-Novelty:** 3
**Q2-2 Correctness-Technical Quality:** 3
**Q2-5 Clarity Of Writing:** 4

**Q1 Summary And Contributions:**

The paper considers the classic shortest path problem and presents an approach to learn latent weights/cost associated with routes using trajectory datasets. The main contributions of the work can be summarized as follows:

1. A differentiable implementation, and its analysis, of the well known Floyd-Warshall algorithm to compute probabilistic shortest paths between various source-target pairs.
2. A novel loss function that is based on frequency counting of nodes on shortest paths and their probabilistic equivalent obtained from point 1.
3. Inference of paths on unseen source-target pairs.
4. Empirical analysis of the approach with state of the art approaches.

**Q2-3 Extent To Which Claims Are Supported By Evidence:**

3: Good: the main claims are supported by convincing evidence (in the form of adequate experimental evaluation, proofs, (pseudo-)code, references, assumptions).

**Q2-4 Reproducibility:**

3: Good: key resources (e.g. proofs, code, data) are available and key details (e.g. proofs, experimental setup) are sufficiently well-described for competent researchers to confidently reproduce the main results.

**Q3 Main Strengths:**

I really like the way the authors defined the loss function using the highest index used by a shortest path. I have not seen such a definition before and it appears to be quite novel.

The paper is nicely organized and easy to read. The theoretical analysis is comprehensive and support the key ideas of the paper.

 The experimental analysis, while not on par with the best of the papers, is solid.

**Q4 Main Weakness:**

Well, if I have to be nit-picky I think the paper can be improved upon in the experimental analysis part. Most of the datasets are synthetic/toyish (except perhaps the taxi dataset). I am very curious to understand how this approach would fare on a much larger geospatial dataset (such as country-wide maps viz. google maps). I was also very interested to know how the context vector part of the input is provided (weather conditions, traffic, time of the day, season etc.) but was somewhat let down there. I think determining the context itself is a massive problem in its own right though.

**Q5 Detailed Comments To The Authors:**

As I mentioned the paper is very well written and I have no major comments on that. Could you comment on the context vector (x_n) used in the algorithm and does it feature, in anyway, in the experiments? Also could you comment on your choice of baselines a bit -- for example, you included none of the RL based approaches as baseline.

**Q9 Complying With Reviewing Instructions:**

Yes

---

> ### Author Rebuttal · Authors · 2024-04-05
>
> We would like to thank the reviewer for appreciating our work and for the highlighted comments/questions. Below, we try to answer the comments/questions regarding: I) the contextual feature; ii) the scalability to bigger graphs; iii) the choice of baselines.
>
> Point 1 (Contextual feature)
>
> We have dealt with two types of “x_n” in our experiments. Images (maps) for the Warcraft experiments; and tabular data for the other experiments (time of the day, weekday, holiday, etc..).
> Warcraft experiment (Images as context): the way “x_n” is inputted in the model is straightforward: for paths that are observed in the same map, we select the same image. This means that a single sample (from the DataSP perspective) contains one map (“x_n”), and many paths observed on that map. This means that our algorithm is more effective when the ratio between the number of paths / map is high. While DBCS (and all other baselines) need to be adapted to process these many paths in parallel, we don’t need to do that.
> Other experiments (Tabular as context): Here, we might not have exactly the same “x_n” for different observed paths. For continuous features, like time of the day, this is even impossible. So then our approach consider to group them in $c$ trajectories that have similar contexts (as further detailed in the fourth paragraph of our “Appendix D: Implementation Details”). For the Cabspot experiment, we have notices that the context consideration is somehow powerful. For some of our runs we have observed that the context consideration made the Jaccard Index result increase between 4 and 5 percentage points. This is not a low improvement, considering that it is natural that most of the time the taxi-drivers would aim to take similar routes for the same start/end points.
>
> Point 2 (Scalability to bigger graphs):
>
> We understand that it would be extremely interesting to see our algorithm working for larger graphs. However, we point out that this is not trivial when learning, with context, from many to many start/end nodes. Our method is capable to compete with some recent works that are dealing with approximately the same graph size but limited to single start/end nodes.
>
> Point 3 (Baselines):
>
> To a similar question/comment for other reviewer, we give the same answer below:
> The biggest challenge in adapting other methods to our scenario lies in dealing with many-to-many start and end nodes. At first glance, it might seem simple, but in practice, this leads to scalability issues within existing works. For example, one of our attempts involved using differentiable LP formulations as another baseline, which unfortunately did not yield publishable results due to convergence problems. Additionally, many recent IRL approaches are designed to deal with local features (e.g., features associated with edges), which do not apply to our scenario, necessitating important modifications.
>
> Thank you again for the insightful review!

---

### Official Review · Reviewer_mcs9 · 2024-03-26

**Q2-1 Originality-Novelty:** 3
**Q2-2 Correctness-Technical Quality:** 3
**Q2-5 Clarity Of Writing:** 4

**Q10 Ethical Concerns:**

No.

**Q1 Summary And Contributions:**

This work proposes a differentiable and probabilistic algorithm for solving shortest path problems called DataSP. This method can be seen as a probabilistic version of the Floyd-Warshall Algorithm where neural networks parameterize the distribution of the highest intermediate node between any pair of two nodes. While it has cubic computation complexity, the authors further propose a graph sampling algorithm for excluding some of the nodes during the learning process. Experimental results on both synthetic and real-world experiments are further presented.

**Q2-3 Extent To Which Claims Are Supported By Evidence:**

3: Good: the main claims are supported by convincing evidence (in the form of adequate experimental evaluation, proofs, (pseudo-)code, references, assumptions).

**Q2-4 Reproducibility:**

3: Good: key resources (e.g. proofs, code, data) are available and key details (e.g. proofs, experimental setup) are sufficiently well-described for competent researchers to confidently reproduce the main results.

**Q3 Main Strengths:**

- DataSP's ability to learn from a vast number of trajectories without additional computational cost is a major advantage.
- The authors provide rigorous theoretical guarantees for their approach including its alignment with the maximum entropy principle, which strengthens the method's validity.
- The overall writing is excellent and it provides sufficient background for the authors to understand the proposed method.

**Q4 Main Weakness:**

- The cubic complexity of DataSP seems to be the main concern. Even though graph sampling is proposed to alleviate this issue by excluding some of the nodes during each learning iteration, I wonder how much such exclusions would harm the learning. In the Warcraft map experiment, 100 out of 324 nodes are excluded which is a lot. Is there a way to carry out an ablation study of how the performance is affected as the number of nodes to be excluded increases? Also, is there some heuristics to help choose what nodes to be excluded?
- It seems for the graph sampling method, the node exclusion needs to be applied to the training data. Since the set of nodes to be excluded varies for each learning iteration, does it mean that when such a set is chosen, one needs to do some preprocessing for the training data, and would it be too costly to do so?
- The paper could benefit from a broader comparison with more recent and relevant baselines for predicting shortest paths such as [1].

[1] Ahmed, Kareem, et al. "Semantic probabilistic layers for neuro-symbolic learning." Advances in Neural Information Processing Systems 35 (2022): 29944-29959.

**Q5 Detailed Comments To The Authors:**

See weakness.

**Q9 Complying With Reviewing Instructions:**

Yes

---

> ### Author Rebuttal · Authors · 2024-04-04
>
> We appreciate the reviewers’ comments. We will try to answer the questions regarding the highlighted “main weakness” below:
>
> Point 1: Node exclusion and heuristics
>
> The analysis regarding the number of nodes excluded for the synthetic dataset is provided right before the "Conclusion" section. As inferred from the results in Table 3, even reducing to 20% of the nodes still yields reasonable results, significantly better than the chosen baselines.
>
> Regarding the heuristics for node exclusion, this is also mentioned in the implementation details (Appendix D, last paragraph) as follows: "We select half of the nodes to be interconnected (forming a connected subgraph), and the other half are chosen randomly based on their frequency of appearance in the training data." This heuristic has, in our attempts, slightly outperformed complete random removal of nodes. Furthermore, we value this discussion as it highlights an important avenue for future research: exploring a more general and efficient method for selecting nodes to be excluded in each training iteration.
>
>
>
> Point 2: How expensive is the node exclusion?
>
> Indeed, as the reviewer pointed out, the node exclusion process involves a non-trivial time-consumption aspect. This is because node exclusion is coupled with a local optimization procedure to ensure that the resultant smooth distance between nodes remains consistent, as it would without node exclusion (for the remaining nodes only). However, this local optimization procedure is less time-consuming than the $V^3$ complexity, as it depends solely on the number of edges connected to the selected nodes for exclusion. Therefore, this graph sampling method significantly enhances the overall speed of the process across all our experiments.
>
>
>
> Point 3: Broader comparison
>
> We thank the reviewer for suggesting a relevant paper of which we were previously unaware. We understand that various approaches, such as Neuro-Symbolic in this instance, can be applied to address structured problems. We acknowledge that we did not consider these types of methods in our current work. However, recognizing their relevance, we plan to explore them in future research. Thank you again for bringing this additional direction to our attention.

---

### Official Review · Reviewer_K6WM · 2024-03-30

**Q2-1 Originality-Novelty:** 3
**Q2-2 Correctness-Technical Quality:** 3
**Q2-5 Clarity Of Writing:** 3

**Q1 Summary And Contributions:**

The paper deals with learning the latent cost of transitions on graphs from trajectory demonstrations. Existing works usually rely on heavy simplifications or suffer from poor scalability. This paper proposes to differentiate through the Floyd-Warshall (FW) algorithm so that it can learn from a large volume of trajectories. To differentiate through the FW, the paper leverages smooth max and argmax operators. The paper also proves that the total cost of the generated trajectories follows the maximum entropy principle. In the experiments, the paper shows that the proposed method outperforms the SOTA combinatorial solver and classical machine learning method in predicting paths on graphs.

**Q2-3 Extent To Which Claims Are Supported By Evidence:**

2: Fair: the main claims are somewhat supported by evidence (but the experimental evaluation may be weak, or does not match entirely with the claims, important baselines may be missing, proofs contain important ideas but lack rigor, algorithmic details are only discussed superficially, references are imprecise, assumptions are not sufficiently motivated or explicated, etc.).

**Q2-4 Reproducibility:**

3: Good: key resources (e.g. proofs, code, data) are available and key details (e.g. proofs, experimental setup) are sufficiently well-described for competent researchers to confidently reproduce the main results.

**Q3 Main Strengths:**

The paper is written clearly. The paper proposes a new method to differentiate through FW which can benefit a lot of planning tasks on graph. It is also good to see that the generated trajectories follow the maximum entropy principle.

**Q4 Main Weakness:**

The experiments sections are relatively weak. The paper only compares with some simple baselines. How does the method compare with advanced inverse reinforcement learning (IRL) methods or generative model-based behavioral cloning methods? In the introduction, the paper claims that IRL can only assume a linear cost function. However, this is not true. A lot of deep IRL methods can handle complex cost functions.

Most of the experiments are based on synthetic environments or datasets which limits its practical impact. It would be better if the author could show more real-world applications.

The paper claims that one issue to differentiate through FW is FW can only provide an exact solution, implying an expectation for demonstrations to be optimal. Based on my understanding, the paper addresses this issue by relaxing the solution. However, I am not convinced that this can mitigate the suboptimality of the demonstration unless you can show that the error direction of the demonstrations and relaxation algorithm is the same otherwise you may also accumulate the errors.

**Q5 Detailed Comments To The Authors:**

It would be better if the paper could show the performance of more advanced IRL or behavior cloning methods and more real-world applications.

**Q9 Complying With Reviewing Instructions:**

Yes

---

> ### Author Rebuttal · Authors · 2024-04-04
>
> We are grateful for the reviewer comments, and we will do our best to answer the “main weakness points” highlighted by the reviewer below:
>
>
>
> Points 1 and 2 (Experiments):
>
> We acknowledge that more recent IRL methods, such as Deep IRL, address nonlinear costs. We have detailed this fact in the “Related Work” subsection (first and second paragraphs).
>
> The biggest challenge in adapting other methods to our scenario lies in dealing with many-to-many start and end nodes. At first glance, it might seem simple, but in practice, this leads to scalability issues within existing works. For example, one of our attempts involved using differentiable LP formulations as another baseline, which unfortunately did not yield publishable results due to convergence problems. Additionally, many recent IRL approaches are designed to deal with local features (e.g., features associated with edges), which do not apply to our scenario, necessitating important modifications.
>
> One goal of our experiments was to demonstrate that our approach scales effectively with the variety (different start/target points) and size of the data, which we highlight as our second contribution. This is observed both in the Warcraft and Synthetic problems, with their varying sizes and properties, underlining the importance of these experiments to "support the claims with evidence." Notably, one of our experiments applied to a real dataset (Cabspot) also demonstrates our method's capability in handling the complexity of taxi drivers' decisions.
>
> We acknowledge that it would be beneficial to show that our method can work in bigger graphs (if this is what is meant by “real-world applications”). However, our main objective here was to introduce a method that scales better with the data (compared to other works). For recent works that deal with global context and neural networks with differentiable programming (e.g., [1, 2, 3]), we are addressing problems with approximately the same graph size, but dealing with more complex data properties.
>
>
>
> Point 3 (Learning from path demonstrations):
>
> We recognize that the FW provides the exact solution. However, our smooth version of FW ends up in the possibility of providing a distribution of paths. And we prove that this distribution follows the Maximum Entropy Principle.
>
> Although the proof is under the sampling algorithm (Algorithm 4), it is inferred that the forward pass of DataSP output ($P$) encodes this principle in its values from many to many nodes. This is due to the fact that Theorem 3 is a consequence of the analytical value of $P$.
>
> This means that if the path demonstrations follow an exponential distribution, such that the probability of taking the path is proportional to $e^(-f)$, where $f$ is the path cost, then the “error” of the suboptimal demonstrations would be aligned to the model.
>
> We acknowledge that it might be that the demonstrations do not follow this exponential distribution, thereby lending weight to your critique. Nonetheless, it is worth mentioning that assuming certain properties about demonstrations is a common strategy across the domain. Additionally, this specific assumption is also consider over IRL methods (e.g., [4]).
>
>
> [1] Vlastelica, Marin, et al. "Differentiation of blackbox combinatorial solvers." arXiv preprint arXiv:1912.02175 (2019).
>
> [2] Berthet, Quentin, et al. "Learning with differentiable pertubed optimizers." Advances in neural information processing systems 33 (2020).
>
> [3] Cristian, Rares, et al. "End-to-end learning for optimization via constraint-enforcing approximators." Proceedings of the AAAI Conference on Artificial Intelligence. 2023.
>
> [4] Ziebart, Brian D., et al. "Navigate like a cabbie: Probabilistic reasoning from observed context-aware behavior." Proceedings of the 10th international conference on Ubiquitous computing. 2008.

---

### Meta-Review · Area_Chair_MmpH · 2024-04-19

The authors developed a differentiable optimization approach through Floyd-Warshall algorithm for all-to-all shortest path learning by replacing max-operator with softmax, which also incurs cubic complexity. Although making searching algorithm differentiable is not novel and has been investigated in ML community intensively, most of the reviewers provide positive feedbacks.

However, the major concern raised by all reviewers lies in the experiment section: 1), the baseline selection is limited and the inverse RL and behavior cloning are ignored; 2), most of the comparisons are conducted on synthetic environments or datasets, which limits its practical impact.